# Experiences of Dutch Midwives Regarding the Quality of Care during the COVID-19 Pandemic

**DOI:** 10.3390/healthcare10020304

**Published:** 2022-02-05

**Authors:** Roos Hijdra, Wim Rutten, Jessica Gubbels

**Affiliations:** 1Department of Health Promotion, NUTRIM School of Nutrition and Translational Research in Metabolism, Maastricht University, P.O. Box 616, 6200 MD Maastricht, The Netherlands; 2Zorggroep Verloskunde ZuidOost Brabant, Zandberglaan 29, 4818 GH Breda, The Netherlands; wpfrutten@outlook.com (W.R.); jessica.gubbels@maastrichtuniversity.nl (J.G.)

**Keywords:** midwifery care, obstetric care, COVID-19, pandemic, value-based healthcare, pregnancy, quality of care

## Abstract

This study assessed how the quality of care during the COVID-19 pandemic has been experienced by Dutch midwives. At the beginning of May 2020, 15 Dutch midwives were interviewed during the first wave of the pandemic. The interviews included questions based on the value-based healthcare framework by Porter. The interviews were transcribed verbatim, coded, and analyzed according to recurrent themes using the directed content analysis approach. Key themes identified included high quality midwifery care, information provision, costs, under/over treatment, interprofessional collaboration, and shared decision making. The quality of midwifery care during the COVID-19 pandemic was experienced to be sufficient, given the challenging circumstances. The midwives experienced the lack of face-to-face check-ups to be problematic. Unclear information and lack of personal protective equipment caused stress and confusion, and they worked an additional 2–4 h per working day. Some pregnant women were hesitant to call or visit them when they thought something was wrong. The midwives perceived some advantages in using video or telephone calls. Considerations for future pandemics include an additional face-to-face check-up between 16 and 27 weeks of pregnancy and one postpartum visit. For post-pandemic care, providing a check-up through telephone or video call could be offered in certain cases.

## 1. Introduction

On the last day of 2019, the emergence of COVID-19 was first reported to the World Health Organization [WHO]. Only a month later, the COVID-19 outbreak was declared an international emergency and halfway through March, it was declared a pandemic [1]. In the Netherlands, the spread of COVID-19 became alarming at the beginning of March 2020. Halfway through May 2020 there were over 44,000 confirmed cases and over 5400 confirmed deaths due to COVID-19 in the Netherlands. Drastic measures, such as social-distancing and working from home, were taken [2]. For the healthcare sector, including obstetric care, most of their regular check-ups had to be cancelled or conducted through video calls or telephone.

Obstetric care in the Netherlands is divided into primary and secondary care [3]. Primary obstetric care is provided by independent midwifery care practices and targets pregnant women with no medical contraindications. Pregnant women within primary care can decide whether they want to give birth to their child at home or at a hospital. Secondary care targets pregnant women who have medical contra-indications, and who are cared for by an obstetrician or midwife in a hospital. However, both types of obstetric care work together very closely because pregnant women often transfer between the two types during their pregnancy [4]. On average, 86–87% of all pregnant women are counselled in primary obstetric care at the start of their pregnancy [5], which is what this study focuses on.

In the first few months of the pandemic, as it was still spreading and evolving, midwives constantly had to adjust their work to simultaneously deal with new measures from the government and the concerns of the pregnant women [6]. Table 1 displays the differences between ‘regular’ primary midwifery care and primary midwifery care during the first two months of the COVID-19 pandemic in the Netherlands. The largest differences in provided care can be seen in the first and second trimester, with fewer check-ups. Additional regulations included partners and children not being allowed to accompany the pregnant woman to check-ups and ultrasounds during the pandemic. Furthermore, during labor, only one other person was allowed to be present (e.g., the expectant father), besides the midwife and the pregnant woman [7]. In addition, while midwives visit the mother and infant at least three times in the first week postpartum during regular care [8], these visits were replaced by telephone check-ups during the COVID-19 pandemic. However, maternity care was still mostly available. In the Netherlands, maternity nurses provide post-partum care at home for at least the first 8 days after birth. Their tasks include looking after the mother and infant, and educating parents on how to take care of the infant and maternal post-partum health, among others [9].

The changes in midwifery care induced by the pandemic may have altered the experiences for pregnant women as well as for midwives delivering the care [7]. At the time, research indicated that there is no increased chance of getting COVID-19 during pregnancy, nor has the virus been identified as easily transmittable from mothers to their fetuses [10]. Unfortunately, it has become clear that COVID-19 during pregnancy can have adverse effects for both the pregnant woman and her child, such as pre-term birth and pre-eclampsia [11]. A recent questionnaire study among Dutch midwives showed that midwives were positive about the decreased number of consultations and the increased confidence in home births. However, they noticed that sometimes the quality of care was compromised [12]. In addition, qualitative studies from Belgium and Australia indicated that the midwives experienced high levels of stress because of the constantly changing measures and lack of resources which negatively affected the quality of care [13,14].

**Table 1 healthcare-10-00304-t001:** Regular midwifery care vs COVID-19 midwifery care in the Netherlands, according to the KNOV (Royal Dutch Midwifery Organization).

Week of Pregnancy	Regular Midwifery Care [15]	Midwifery Care during COVID-19 [7]
6–8	Face-to-face intake	Intake through telephone
8–10	Face-to-face counselling prenatal screening	Counselling through telephone
14–26	2–3 face-to-face check-ups including one ultrasound around 20 weeks, 1 group counselling session on pregnancy	1–2 face-to-face check-ups, potentially including and ultrasound around 20 weeks, depending on whether the ultrasound is performed by the midwife or by an external ultrasound technician
27–40	6–8 face-to-face check-ups, 1 group counselling session on childbirth and the post-partum period	6 face-to-face check-ups
41–42 (if applicable)	1–3 face-to-face check-ups	1–2 face-to-face check-ups
Postpartum	At least 3–4 home-visit check-ups	2–3 check-ups through telephone, video call or a window visit
Total	13–22 face-to-face check-ups	8–10 face-to-face check-ups, 4 telephone check-ups

To provide a Dutch qualitative perspective, this study examined the experiences of Dutch midwives regarding the quality of care during the first two months of the COVID-19 pandemic. This is examined through qualitative semi-structured interviews with midwives who have worked during the first two months of the COVID-19 pandemic.

## 2. Methods

### 2.1. Study Design and Procedures

The current study used a qualitative design, through semi-structured interviews with Dutch primary care midwives. The interviews were not restricted to a fixed set of questions; the midwives were able to elaborate on other topics they felt were important. The interviews were conducted at the beginning of May 2020, two months after the start of the COVID-19 outbreak in the Netherlands. The interviews were performed through telephone or a secured Zoom account, depending on the preferences of the participants. At the beginning of each interview, background information regarding the study was provided. The respondents were then asked to give their verbal informed consent for participation in this study and for audio recording of the interview. The interviews lasted, on average, 44 min, and were audio recorded using the recording software on a smartphone and laptop. The ethical committee of the Faculty of Health, Medicine and Life Sciences of Maastricht University provided their approval for the study (approval number: 6138684).

### 2.2. Participants

Midwives who were members of one of three selected midwifery collaboration cooperation’s in the province of Noord-Brabant were asked to participate in this research. Noord-Brabant was one of the COVID-19 hotspots during the first wave of the pandemic in the Netherlands [7]. The recruitment of participants was done by email: 85 midwives from 19 different midwifery practices were invited to participate in the study. The inclusion criteria were that the midwives were working as a midwife in primary care during the COVID-19 pandemic in the selected region. There were no other inclusion or exclusion criteria. A total of 15 midwives from 13 different midwifery practices agreed to participate and were interviewed. Hence, the participant response rate was 17.6%, covering 68.4% of the approached midwifery practices.

### 2.3. Theoretical Framework and Instrument

The value-based health care framework from Porter was adopted to structure the interviews [16]. Value based healthcare builds on three principles. The first principle is high value health care. This entails that the quality of care should be the most important factor in order to improve the health of the patient. Porter explains that healthcare providers should only provide the care that they excel in, avoiding over or undertreating their patients. Since value-based healthcare is focused on high value health care from the view of the patient, shared-decision making heavily influences the quality of care for pregnant women. Secondly, healthcare should be focused over the entire process of care, not just one treatment. This emphasizes interprofessional collaboration of independent midwifery practices, hospitals, and maternity care. Lastly, the need to measure value. Within the value-based healthcare framework, this entails that the outcomes and costs of care need to be measured over the entire care for a medical condition [16]. Based on the value-based health care framework an interview guide was developed (see Table 2), discussing the quality of midwifery care during the COVID-19 pandemic, compared to delivery of care before the pandemic. The focus on a high quality of care was translated to whether the midwives have had the resources to provide high-quality care and how they have experienced providing care during the pandemic. The interview guide was used to structure the interviews.

### 2.4. Analysis

After the interview, the audio recordings were fully transcribed verbatim and subsequently coded. The coding scheme was created based on the questions of the interview guide, the underlying theoretical framework (described above) and additional topics that were mentioned during the interviews. NVivo 12 was used to facilitate the coding [17]. A directed content analysis approach was used to assist in coding [18]. Hence, codes were created based on the theoretical framework described above (Porter, 2008) and recurring themes from the interviews. The interviews were coded by one author. Doubts or unclarities were discussed within the team. After coding the interviews, the codes were individually analyzed to structure recurring themes and differences.

## 3. Findings

The findings are structured around the main themes identified during the interviews: background, high quality midwifery care, information provision, costs, under/over treatment, interprofessional collaboration, and shared decision making. Table 3 summarizes the background information of the participant. The table depicts that most participating midwives were working full-time. Additionally, only one midwife had underlying health issues which could increase the severity of a COVID-19 infection. All midwives did not care for or live with anyone who was subjected to such issues.

### 3.1. Experiences of Midwives on Providing Care during the COVID-19 Pandemic

At the beginning of the pandemic, the midwives experienced difficulties in keeping up with all the new measures and guidelines that were constantly published. This took up a lot of time, which was experienced as exhausting by some of the midwives. Midwives and the pregnant women felt agitated and were afraid of what was to come. Moreover, midwives had to trust the pregnant women to not come to their practice when experiencing COVID-19-like symptoms. Some pregnant women did, however, come to the practice with symptoms.
“*I think that the biggest obstacle for me was that I constantly had to think about everything. I’m making decisions differently now. Before, you would always say: ‘oh just come by our practice’ or ‘I’ll come visit you’ [if a client experienced any problems]. Now I must remember to first ask whether they are showing any symptoms, then I have to reconsider what I have to do.*”(Respondent 2)

Additionally, some midwives mentioned that it was difficult to get all the midwives in their practice in line with each other, due to the different interpretations of information they received, as well as differences in opinions. After a few weeks, when the regulations became clearer and the midwives got more accustomed to the situation, they experienced it to be somewhat easier.
“*What I liked is that I supported all of the measures that were taken. This made it easier to communicate them to our clients. And I have to say, I was surprised about how much is still possible. And how happy I was with businesses such as Zoom,… Imagine if this had happened 50 years ago, it would have been a much bigger problem.*”(Respondent 11)

However, almost all midwives mentioned that they were not fond of the situation nor how it influenced their profession. Most of the midwives found it difficult that they were not able to provide the quality of care that they strived for during this period. Nevertheless, they attempted to provide the best care possible.
“*It’s just not fun anymore. You want to give people an ultrasound or give them a shoulder to cry on, but you always have to think about keeping distance and the safety measures*.”(Respondent 4)

The midwives were not scared of getting the virus themselves, even though they were in a region where, relatively, the most COVID-19 cases in the Netherlands occurred at the time. Nevertheless, they were fearful regarding the potential risk of spreading COVID-19 onto pregnant women or family members.
“*You do your best to prevent it [spreading of the virus]. If it happens despite this, then you could not have prevented it. But I still get afraid when I hear of cases where young people get really ill from it. And for yourself it would be horrible of course, but it is worse for the young children you would leave behind. At the beginning, I experienced anxiety often, when you entered someone’s home and thought: was this a moment [of infection/spreading] or not?*”(Respondent 1)

### 3.2. Experiences on Information Provision on COVID-19

The midwives mentioned that they used the websites and newsletters from the Royal Dutch Organization for Midwives [KNOV] and the Dutch National Institute for Health and Environment [RIVM] for national information. For regional information regarding midwifery care and COVID-19, they used the information provided by the regional ‘Corona Taskforce’. This Taskforce was established by the KNOV to organize midwifery care at a regional level during the COVID-19 pandemic. This taskforce included important stakeholders such as the chairwomen from regional midwifery collaborations. However, the information provided by the KNOV and the Taskforce was perceived as unclear on several occasions. Several participants reported that the information, especially on personal protective equipment [PPE], could be interpreted in multiple ways, which made it difficult to implement. This created differences between midwives, midwifery practices, and hospitals. Additionally, there was a severe shortage of PPE at the beginning of the pandemic. This all caused a lot of stress for the midwives.
“*We just don’t know what the best thing to do is. Everybody wants to do it well, but exaggerating and putting on too much protective equipment is also not good, because that only causes fear.*”(Respondent 7)

At the beginning of the pandemic, new information from the KNOV regarding COVID-19 frequently became available and documents were often revised and updated. This made it difficult for the midwives to maintain overview and keep track of the most updated information. Most of the information was provided on time. However, the first time a relaxation of the national COVID-19 measures was introduced by the KNOV, the measures were published late and leaked to the press before official publication. This caused a lot of confusion and frustration among both midwives and pregnant women. It also resulted in a lot of additional work for the midwives, because the regional measures they had to adhere to were now stricter than the reduced national measures.

Some of the respondents indicated that they would have appreciated more general information from the RIVM for pregnant women on the effects of COVID-19 during pregnancy. However, other respondents mentioned that because of the differences between the current region and national regulations, it was better to keep the dissemination of information solely with the midwives.

### 3.3. Interprofessional Collaboration

The only thing that changed in the interprofessional collaboration between midwives and the obstetrical care in the hospital during the COVID-19 pandemic, was that most of the meetings were held online instead of face-to-face. The quality of the collaboration was deemed to be the same, however. Collaboration with maternity home care nurses was also satisfactory. At the beginning of the COVID-19 pandemic, the midwives no longer visited pregnant women and newborns in the first week postpartum. Consequently, most of the responsibility shifted to the maternity nurses. The midwives were of the opinion that the maternity nurses sufficiently managed the situation.
“*The collaboration with the maternity nurses went fine. I did realize I think, that we but a pretty large burden on the shoulders of the maternity nurses, because they are our eyes and ears now, even though they are not trained for that. Normally, we are able to decide together: is this child looking a bit yellow? These things are a matter of judgement, which now falls fully on the shoulders of the maternity nurse.*”(Respondent 11)

### 3.4. Costs

The respondents reported that, on average, they worked 2–4 h more per day because of the COVID-19 measures. This lasted for about a month, after which they reported that they slowly became accustomed to the measures. These additional hours mostly resulted from the midwives having to call all their clients before their face-to-face check-ups, to triage them and to inform them about new measures. Further additional activities included the telephone check-ups and having to keep up with all the changes in regulations and communicating these to pregnant women and other stakeholders. The additional work did depend on whether the midwife owned a midwifery practice or not. The owners felt more responsible and handled more of the administrative tasks, such as converting new measures to the implications for their specific practice. Some respondents mentioned that this was starting to take a toll on their mental health, whilst others did not experience such a burden. Additional costs due to COVID-19, besides the increase in working hours, mainly consisted of costs for PPE and increased telephone bills.
“*It is funny, because you would think that a telephone consult would be faster, but it takes a lot more time. Normally, you can explain thinks using your computer or a folder, but now you have to explain it all by telephone. After the telephone call, we then have to deliver said folders to the clients. So in that case it costs a lot more time, on average about two to three hours more I think.*”(Respondent 12)

### 3.5. Under/Overtreatment

Although the respondents mentioned that they were able to comply with the prescribed KNOV schedule [see Table 1], they did report a need to call the pregnant women more frequently than prescribed by the KNOV. There was as specific need for increased check-ups between 16 and 27 weeks of pregnancy, as this was a relatively long period without a scheduled check-up. This extensive time between check-ups was found to be mentally challenging for the pregnant women. Furthermore, the first week postpartum without a face-to-face check-up was experienced to be difficult by midwives, and they noticed that this was also difficult for the pregnant women and maternity care nurses. It was difficult, for example, to see how the stitches were healing or whether the baby was looking healthy through video calls. Opinions on these video or telephone check-ups varied. Most midwives preferred telephone calls; they thought this was easier than video calls. Some midwives were pleased with the alternatives for face-to-face check-ups. Sometimes it was easier to plan such check-ups by telephone of video calls, although it did not necessarily save much time. Other midwives mentioned that they missed the non-verbal communication through telephone or video call.
“*Telephone check-ups are more difficult, that really is a shame. You cannot see them, and if they say: ‘My back hurts’, they cannot point it out over the telephone. And when they come to the face-to-face check-up, it still takes just as long, so there is no time won there.*”(Respondent 13)

When a pregnant women needed (face-to-face) care, but this was not in the KNOV schedule, the midwives would provide it anyway. However, this meant that only the women who made clear that they needed or wanted additional care, received the care. Some midwives mentioned that they were afraid that they overlooked some subtle additional needs from clients, which they would normally only notice during a face-to-face check-up through the women’s nonverbal communication.

Midwives perceived the COVID-19 midwifery schedule to diminish the quality of care they provided compared to before the COVID-19 pandemic. Partners and children were not allowed to come to face-to-face check-ups, therefore, it was difficult to get a clear overview of the family circumstances. Additionally, pregnant women did not always know all the midwives working at the practice because there were fewer check-ups. As a result, occasionally, a midwife who the pregnant woman did not know assisted during labor, which was deemed undesirable. Before the pandemic, pregnant women often got to meet all midwives working at the practice before their delivery. However, overall, they believed the quality of care they provided was sufficient, considering the challenging circumstances.

The midwives made the following recommendations on what changes they would like to continue after the pandemic. They mentioned that multidisciplinary meetings with the hospitals, other organizational meetings, the intake consultation with clients, and some of the face-to-face check-ups during pregnancy or postpartum could be replaced by telephone or video call check-ups. However, some of the midwives indicated they would like to keep all contact moments with their clients face-to-face.

### 3.6. Experiences Pregnant Women

The respondents all mentioned that most pregnant women accepted the schedule changes. Some pregnant women did mention to their midwife that the decreased number of ultrasounds and the absence of their partner accompanying them to the face-to-face check-ups was difficult for them. Almost all the respondents mentioned that there were a few pregnant women who did not take the changes in care well, or that some partners continued to come at appointments despite the requests not to come.
“*There were clients who were making a fuss about wanting an ultrasound, although it was not medically necessary. They were asking when it would be necessary to get an ultrasound and I explained it to them. The next day, they would call me with those exact symptoms for an ultrasound. Then you have no other option but to give them an ultrasound.*”(Respondent 2)

There were a few pregnant women that were too afraid to visit the midwife. They were afraid of getting infected during the visit, despite midwives making clear that they were taking all necessary precautions. Some midwives even noticed that some pregnant women were too afraid to call the midwife when something was wrong. They were afraid of burdening the midwives or having to visit the hospital. Simultaneously, obstetric caregivers in the hospitals were noticing that fewer pregnant women were referred to secondary obstetric care. The midwives indicated that this observation scared them, because they had not noticed pregnant women who needed to visit the hospital but did not visit, nor that they were referring people less frequently to secondary care in comparison to before the COVID-19 pandemic.
“*What we noticed in the region is that we had to beware of still caring for the people who really need it the most. I got scared by an example where they [the midwife] had missed something [in a case of intrauterine foetal death]. What kind of symptoms did the pregnant woman have and why was she not seen? … in the beginning we had to hold everybody off as much as possible. But … If someone calls with severe worries and symptoms, then you will still go and visit them. If they are very worried, you simply cannot let them down.*”(Respondent 1)

### 3.7. Shared Decision Making

There was less room for shared-decision making during the COVID-19 pandemic. Due to the COVID-19 measures and guidelines, it was not always possible to follow the wishes of the pregnant women. Additionally, the telephone or video call check-ups made it more difficult to fully explain their options and have a discussion together.
“*You say more often: ‘This is necessary now’. There is less room to talk about it and decide about it together. It is of importance for the general health and wellbeing, and most people get that. There is less wiggle room to divert from those measures.*”(Respondent 14)

## 4. Discussion

This qualitative study assessed the experiences of midwives regarding the quality of care in the Netherlands during the COVID-19 pandemic. Five main themes can be identified from the findings: the increased threshold of contacting the midwife by pregnant women when something is wrong; the shortage and unclear information on PPE; miscommunication on new measures; a decrease in shared-decision making; and the quality of midwifery care.

The midwives noticed that there were fewer pregnant women contacting them for care. In line with this, all three hospitals providing secondary obstetric care in the region indicated that fewer pregnant women were referred to the hospital. In Italy, a similar situation occurred where hospital statistics showed a significant reduction of 80% in pediatric emergency department visits and family pediatrician visits during the COVID-19 pandemic. They concluded that this decrease is likely due to a scarcity of available resources and a fear of exposure to COVID-19 [19]. In the United States, this fear of infection at the hospital was reflected in an increased demand for out-of-hospital births [20]. In India, delivery units at the hospital also noticed a decrease in hospital deliveries [21]. Furthermore, pregnant women in Kenya and Ghana mentioned that they were afraid of visiting the midwife or the hospital in fear of getting infected with COVID-19, after the number of visits to healthcare professionals was found to decrease [22,23]. This is in line with the observations of the KNOV [24]. Midwives in Australia saw clients less, or mostly through video- or phone calls, which caused anxiety and large concerns for the midwives for missing critical signs [25]. A quantitative study with Dutch midwives working both in primary and secondary obstetric care, showed that primary care midwives referred clients less often to the hospital. The secondary care midwives also had fewer consultations for decreased fetal movements. Additionally, they noticed an increase in homebirths, which they speculated was also due to a fear of getting infected at the hospital [12]. This anxiety and worries about their client’s health, together with their increased working hours, was starting to take a toll on some of the midwifes’ health that were interviewed in this study. In the long-term, this could potentially lead to worse health issues, such as burnout [26]. More research on the long-term consequences for midwives is urgently needed to further examine this.

Furthermore, the midwives experienced a shortage on PPE and unclarity of PPE measures. During a previous influenza pandemic in the Netherlands, a similar situation occurred, where the measures on PPE were unclear to general practitioners (GPs) [27]. The GPs mentioned that the lack of clarity regarding the measures made it difficult to implement them into their daily work. This caused almost half of them to disregard the measures, mainly because the GPs were not aware of the patients’ infection status [27]. A literature review on the use of PPE during pandemics only indicated which type of PPE should be used when dealing with a COVID-19 case without specifying which PPE to use for a healthcare worker who is dealing with someone not suspected of being infected with COVID-19 [28]. This is in line with the current study, where the midwives had trouble deciding when to use which type of PPE. The measures were clear when there was a confirmed case of COVID-19, but when someone showed no potential symptoms, it was not clear whether they should still wear PPE. Besides the issues with unclarity of PPE measures, shortages were experienced around the world, which caused anxiety among the healthcare workers [29,30,31]. Further concern was caused by the difficulty to read facial expressions and clear communication whilst wearing PPE [32].

The participants indicated that there was an occurrence of severe miscommunication on new measures. The midwives communicated clearly to the KNOV that this put them in a difficult situation, so the KNOV took action. The next time that new measures were introduced, midwives were invited to provide feedback on the concept measures. In addition, the KNOV sent the final document to the midwives more than two weeks before implementation. This gave the midwives time to organize the implementation of the new measures, on a regional and individual level [33]. The importance of clear and timely communication between health care organizations and personnel is also supported by the Pandemic Influenza Response Plan and Strategic Framework of Public Health England, based on previous pandemics [34]. Asking involved professionals for feedback on new measures before finalization is thus also advised for the measures on PPE.

Shared-decision making, which is an important part of value-based health care [16], decreased during the COVID-19 pandemic. Previous studies on pandemics and their ethical challenges concluded that ethical decision making is often overlooked under special circumstances such as pandemics [25,35]. The limited room for pregnant women’s choices due to national regulations made it difficult to offer full shared-decision making, because some options were not available anymore [25]. Smith and Silva argue that ethics should be incorporated into the measures that are taken during a pandemic, and that under special circumstances, there is room for different decisions [36]. For midwives this could, for example, mean that if a pregnant woman has experienced a previous traumatic labor, there is room to talk about having an extra person present during labor. Video calls instead of phone calls could also be a way to have more personalized contact with pregnant women [37,38], facilitating shared-decision making. Additionally, some measures can be interpreted in many ways and can give slightly different choices to the pregnant women. Differing personal values of each midwife and midwifery practice also influence their perspective [39]. With more attention to ethics in pandemic measures, it could be possible to maintain a certain level of the shared decision making.

Lastly, considering the circumstances, the overall quality of care during the COVID-19 pandemic was still considered as sufficient by the midwives. In case of a new wave of COVID-19, or a new pandemic, the KNOV COVID-19 schedule was expected to be sufficient, pending a few changes to be made. Firstly, a check-up through telephone or video call should be scheduled in the current 16 to 27-week gap, around 21 or 22 weeks of pregnancy. The midwife can decide when to schedule this herself, depending on whether they perform the 20-week ultrasound themselves and therefore already see the pregnant woman, or this ultrasound is performed elsewhere (e.g., a hospital). Secondly, a face-to-face check-up in the first week postpartum is advised, around day three or four postpartum. Additionally, clear guidelines with multiple examples should be created for the use of PPE, since this is essential when dealing with COVID-19 patients [40,41].

For the post-pandemic midwifery care, certain check-ups could be offered over telephone or video call. However, this should only be used if a face-to-face physical check-up is not necessary. This could be the case for the intake appointment, and some of the check-ups during pregnancy and postpartum. A previous study showed that patients sometimes prefer a telephone check-up over a face-to-face check-up. NHS England provided the choice between a face-to-face consult or a telephone consult for a flue triage, and 60% of the respondents chose for the latter. This can also release some of the stress on the healthcare system [42]. During the COVID-19 pandemic, chiropractors in the United States also started using telephone and video call check-ups. They were able to provide high quality care over telephone or video call and the patient satisfaction was high [43]. Additionally, some of the organizational and multidisciplinary meetings that would normally happen face-to-face can be held through telephone or video call.

Further research is still necessary in order to determine what the effects of the COVID-19 pandemic are on the health and wellbeing of pregnant women and their offspring, both regarding the effects of the changes of care, as well as regarding the effects of the virus itself. There are some first indications that COVID-19 can have adverse effects for both the pregnant woman and the child [11]. Additionally, the experiences and perspectives of pregnant women and other stakeholders in the midwifery care sector, such as obstetricians and maternity care nurses, should be investigated in future studies. This can provide further indications on how to handle a future outbreak and optimize regular obstetric care.

### Strengths and Limitations

One of the limitations of this study is that the study was performed during the very quickly changing circumstances of the first wave of the COVID-19 pandemic in the Netherlands. As a result, the interviews were each held at slightly different stages of the pandemic, with different national and regional measures and regulations. In addition, the interviews were performed through Zoom or telephone, due to the COVID-19 measures. This made it difficult to pick up on nonverbal communication, although it did seem to lower the participant burden, as the respondents were able to participate from home at a moment they preferred. Additionally, the interviews were performed in only one region of the Netherlands, which had one of the highest number of COVID-19 cases in the Netherlands at that time [44]. Therefore, the generalizability to other parts of the Netherlands or other countries might be limited. However, the participating midwives were certainly exposed to the implications of the COVID-19 pandemic due to this. Furthermore, this study did not include the views of the pregnant women. The study sample of 15 midwives can be considered as limited, although it did represent 68% of the midwifery practices in the selected region and data satiation was reached. The sample was diverse, with midwives working for only a year to over 30 years of experience and working in very small to large midwifery practices. This study is, to our knowledge, the first qualitative study to report on the quality of obstetric care by midwifes during the COVID-19 pandemic from the viewpoint of Dutch midwives. The interviews were conducted during the first few weeks of the COVID-19 pandemic, which offers a clear insight on how the midwives dealt with the uncertainties at the time. This is useful information for preparing for potential future pandemics.

To conclude, the quality of care during the COVID-19 pandemic was experienced to be sufficient, considering the circumstances. For future obstetric care during a pandemic, an extra telephone check-up and a face-to-face visit postpartum are recommended to increase the quality of care. After the pandemic, the midwifery care schedule is advised go back to the way it was organized before the pandemic. However, telephone or video call check-ups can be offered to the pregnant woman when a face-to-face check-up is not necessary. Video calls can also be used more often for organizational meetings in the future.

## Figures and Tables

**Table 2 healthcare-10-00304-t002:** Interview guide on the experiences of midwives during the COVID-19 pandemic regarding the quality of care.

Theory/Concept	Question
Background	Have you completed any other higher education studies besides Midwifery? If so, which study?
	For how long have you been working as a midwife?
	At which midwifery practice do you work?
	For how long have you been working at this specific midwifery practice?
	How many hours do you normally work a week? [part-time/full-time]
	Do you have any underlying health conditions that may increase the severity of a COVID-19 infection?
	Do you live with or care for people who have underlying health issues that may increase the severity of a COVID-19 infection?
High-quality midwifery care	How do you experience providing midwifery care during the COVID-19 pandemic? Is it more or less challenging than you expected? [mentally and physically]
	What is your perception on how pregnant women and their partners experience the quality of midwifery care during the COVID-19 pandemic?
	What do you think of the way that midwifery care is organized at the moment?
	What do you think could increase the quality of midwifery care given the current circumstances?
	What are, in your opinion, the biggest obstacles in currently providing midwifery care?
	Do you notice any effects of social media coverings of COVID-19 on you and your clients?
	Which of the changes because of COVID-19 could be continued after the pandemic has ended?
Information provision	Where do you primarily obtain information from regarding midwifery care and COVID-19?
	Was/is this information easy to find and has it been published on time?
	Was this information comprehensible?
	Has it been clear when to use what personal protective equipment?
	Has it been clear when you and your pregnant women should get tested for COVID-19?
	Were new measures and/or guidelines clear?
	Do you think that large organizations such as the KNOV and RIVM could have done more for the pregnant women with regards to information provision?
Costs	Do you spend more or fewer hours working because of the changes due to COVID-19? How many hours? What caused this increase or decrease?
	Do you think the extra costs you made because of COVID-19 and/or the extra hours you worked will be reimbursed?
Under/over treatment	How often did you have to deviate from the COVID-19 midwifery care schedule provided by the KNOV?
	Do you feel like you had enough contact with your clients during pregnancy or after childbirth?
Shared decision making	To what extent were you able to practice shared decision making?
Interprofessional collaboration	What do you think of the interprofessional collaboration during the COVID-19 pandemic?

Abbreviations: KNOV = Royal Dutch Organization for Midwives, RIVM = National Institute for Public Health and the Environment.

**Table 3 healthcare-10-00304-t003:** Background information participants.

Participant	Total Working Years as Midwife	Working Years at Current Midwifery Practice	Working Part-Time/Full-Time	High Risk Group ^a^	Care for or Live with High Risk Group ^a^
1	14	13	Full-time	No	No
2	16	14	Full-time	No	No
3	1.5	1	Full-time	No	No
4	12	8	Full-time	No	No
5	3	2	Full-time	No	No
6	5	0.5	Part-time	No	No
7	16	15	Full-time	No	No
8	5	1.5	Full-time	No	No
9	33	24	Part-time	No	No
10	29	15	Full-time	No	No
11	7	3.5	Full-time	No	No
12	5	5	Part-time	Yes	No
13	22	12	Full-time	No	No
14	11	11	Full-time	No	No
15	5	1	Full-time	No	No

Notes: ^a^ High risk group = person with underlying health issues which may increase severity of a COVID-19 infection.

## Data Availability

The data presented in this study are available on request from the corresponding author. The data are not publicly available due to privacy reasons.

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
