# Peer review of "Experiences of Dutch Midwives Regarding the Quality of Care during the COVID-19 Pandemic"

_healthcare, 2022, doi:10.3390/healthcare10020304_

Round 1

Reviewer 1 Report

This is a qualitative study about experiences of Dutch midwives regarding the quality of care during the Covid-19 pandemic.

  1. The abstract should state that this was a qualitative study done during the first weeks of the pandemic, the qualitative theory used, and key themes. I encourage the abstract focus on the qualitative theory concepts.
  2. The introduction appears appropriate regarding historical data from the pandemic. This was nicely supplemented with Table 1.
  3. In the methods, please include the qualitative method that you used. You state the theoretical framework (page 4), but not the qualitative method. The qualitative method used is critical because it drives the rest of the study and how findings are presented. You stated the study was not restricted to a set of questions, yet you shared a set of questions in Table 2. Because you focused on the theory, the themes do not appear to evolve from the participants but the predetermined questions for the study.
  4. How were themes determined since coding was based on the interview questions? I question that you had themes. The quotes from participants are not frequently shared. Usually, these quotes are critical in qualitative work.
  5. The discussion begins with 5 main themes, but that is not how the results are presented. The discussion does not seem to follow the results.
  6. I encourage adding a section about what was learned that could influence care or be put in place for future pandemics.
  7. Limitations are shared.
  8. The section on strengths and limitations is primarily limitations.

References appear recent, but there are few on qualitative methods.

This is a qualitative study about experiences of Dutch midwives regarding the quality of care during the Covid-19 pandemic.

  1. The abstract should state that this was a qualitative study done during the first weeks of the pandemic, the qualitative theory used, and key themes. I encourage the abstract focus on the qualitative theory concepts.
  2. The introduction appears appropriate regarding historical data from the pandemic. This was nicely supplemented with Table 1.
  3. In the methods, please include the qualitative method that you used. You state the theoretical framework (page 4), but not the qualitative method. The qualitative method used is critical because it drives the rest of the study and how findings are presented. You stated the study was not restricted to a set of questions, yet you shared a set of questions in Table 2. Because you focused on the theory, the themes do not appear to evolve from the participants but the predetermined questions for the study.
  4. How were themes determined since coding was based on the interview questions? I question that you had themes. The quotes from participants are not frequently shared. Usually, these quotes are critical in qualitative work.
  5. The discussion begins with 5 main themes, but that is not how the results are presented. The discussion does not seem to follow the results.
  6. I encourage adding a section about what was learned that could influence care or be put in place for future pandemics.
  7. Limitations are shared.
  8. The section on strengths and limitations is primarily limitations.

References appear recent, but there are few on qualitative methods.

This is a qualitative study about experiences of Dutch midwives regarding the quality of care during the Covid-19 pandemic.

  1. The abstract should state that this was a qualitative study done during the first weeks of the pandemic, the qualitative theory used, and key themes. I encourage the abstract focus on the qualitative theory concepts.
  2. The introduction appears appropriate regarding historical data from the pandemic. This was nicely supplemented with Table 1.
  3. In the methods, please include the qualitative method that you used. You state the theoretical framework (page 4), but not the qualitative method. The qualitative method used is critical because it drives the rest of the study and how findings are presented. You stated the study was not restricted to a set of questions, yet you shared a set of questions in Table 2. Because you focused on the theory, the themes do not appear to evolve from the participants but the predetermined questions for the study.
  4. How were themes determined since coding was based on the interview questions? I question that you had themes. The quotes from participants are not frequently shared. Usually, these quotes are critical in qualitative work.
  5. The discussion begins with 5 main themes, but that is not how the results are presented. The discussion does not seem to follow the results.
  6. I encourage adding a section about what was learned that could influence care or be put in place for future pandemics.
  7. Limitations are shared.
  8. The section on strengths and limitations is primarily limitations.

References appear recent, but there are few on qualitative methods.

This is a qualitative study about experiences of Dutch midwives regarding the quality of care during the Covid-19 pandemic.

  1. The abstract should state that this was a qualitative study done during the first weeks of the pandemic, the qualitative theory used, and key themes. I encourage the abstract focus on the qualitative theory concepts.
  2. The introduction appears appropriate regarding historical data from the pandemic. This was nicely supplemented with Table 1.
  3. In the methods, please include the qualitative method that you used. You state the theoretical framework (page 4), but not the qualitative method. The qualitative method used is critical because it drives the rest of the study and how findings are presented. You stated the study was not restricted to a set of questions, yet you shared a set of questions in Table 2. Because you focused on the theory, the themes do not appear to evolve from the participants but the predetermined questions for the study.
  4. How were themes determined since coding was based on the interview questions? I question that you had themes. The quotes from participants are not frequently shared. Usually, these quotes are critical in qualitative work.
  5. The discussion begins with 5 main themes, but that is not how the results are presented. The discussion does not seem to follow the results.
  6. I encourage adding a section about what was learned that could influence care or be put in place for future pandemics.
  7. Limitations are shared.
  8. The section on strengths and limitations is primarily limitations.

References appear recent, but there are few on qualitative methods.

This is a qualitative study about experiences of Dutch midwives regarding the quality of care during the Covid-19 pandemic.

  1. The abstract should state that this was a qualitative study done during the first weeks of the pandemic, the qualitative theory used, and key themes. I encourage the abstract focus on the qualitative theory concepts.
  2. The introduction appears appropriate regarding historical data from the pandemic. This was nicely supplemented with Table 1.
  3. In the methods, please include the qualitative method that you used. You state the theoretical framework (page 4), but not the qualitative method. The qualitative method used is critical because it drives the rest of the study and how findings are presented. You stated the study was not restricted to a set of questions, yet you shared a set of questions in Table 2. Because you focused on the theory, the themes do not appear to evolve from the participants but the predetermined questions for the study.
  4. How were themes determined since coding was based on the interview questions? I question that you had themes. The quotes from participants are not frequently shared. Usually, these quotes are critical in qualitative work.
  5. The discussion begins with 5 main themes, but that is not how the results are presented. The discussion does not seem to follow the results.
  6. I encourage adding a section about what was learned that could influence care or be put in place for future pandemics.
  7. Limitations are shared.
  8. The section on strengths and limitations is primarily limitations.

References appear recent, but there are few on qualitative methods.

This is a qualitative study about experiences of Dutch midwives regarding the quality of care during the Covid-19 pandemic.

  1. The abstract should state that this was a qualitative study done during the first weeks of the pandemic, the qualitative theory used, and key themes. I encourage the abstract focus on the qualitative theory concepts.
  2. The introduction appears appropriate regarding historical data from the pandemic. This was nicely supplemented with Table 1.
  3. In the methods, please include the qualitative method that you used. You state the theoretical framework (page 4), but not the qualitative method. The qualitative method used is critical because it drives the rest of the study and how findings are presented. You stated the study was not restricted to a set of questions, yet you shared a set of questions in Table 2. Because you focused on the theory, the themes do not appear to evolve from the participants but the predetermined questions for the study.
  4. How were themes determined since coding was based on the interview questions? I question that you had themes. The quotes from participants are not frequently shared. Usually, these quotes are critical in qualitative work.
  5. The discussion begins with 5 main themes, but that is not how the results are presented. The discussion does not seem to follow the results.
  6. I encourage adding a section about what was learned that could influence care or be put in place for future pandemics.
  7. Limitations are shared.
  8. The section on strengths and limitations is primarily limitations.

References appear recent, but there are few on qualitative methods.

Reviewer 2 Report

This paper discussed the experiences of Dutch Midwives regarding the quality of care during the pandemic using the interviews with 15 midwives during the first two months of the Covid-19 break in Dutch. The interviews were well analyzed and examined in terms of face-to-face check-ups experience from the midwives, costs, quality of care, and challenges. The authors concluded additional face-to-face check-ups need to be considered for 16-27 weeks of pregnancy and postpartum visit to combat future pandemic and check-ups through telephone or video call for certain cases may be needed for post-pandemic care. Overall, this paper is well-written; the idea is novel, interesting, and timely considering the challenges for health care workers and pregnant women; the structure is clear. I have a few comments listed here:

Line 37, what does RIVM represent in the first appearance?

Line 51, what does KNOV represent in the first appearance?

Lines 65 and 79, miss a dot at the end.

In the methods section, the interviews conducted at the beginning of May 2020 were interviewing midwives who provided the obstetric care at the first month of Covid-19? Or the first two months of Covid-19? If one month, change “first months” in line 95 and line 97 to “first month”, if two months, add two months accordingly.

Line 122, why the participant response rate was 17.6%, not 100% from the 15 participated midwives?

Lines 162-163, the description is not precise, table 3 shows most midwives did not have any underlying health issues, but all of them did not care for or live with anyone who was subjected to such issues (The last column have all values with No), rather than most.

While the experiment design may not be sufficient considering the small samples (N=15) and the interviews only from the midwives rather than including the pregnant women themselves, the authors realized the limitations and discussed them in the last section, which is good. The paper can be improved by collecting the responses from the registered pregnant women who received the care at the same time period.

In the reference section, keep the citation style consistent, especially for those with the first letter capitalized or non-capitalized in the article titles, such as the second, third, eighth, and so on.
